# Determinants of immunisation dropout among children under the age of 2 in Zambézia province, Mozambique: a community-based participatory research study using Photovoice

Jocelyn Powelson [1], Bvudzai Priscilla Magadzire [2], Abel Draiva,[3] Donna Denno,[4] Abdul Ibraimo,[5] Bonifácia Beleza Lucas Benate,[6] Lídia Carlos Jahar,[7] Zélia Marrune,[7] Baltazar Chilundo,[5] Jalilo Ernesto Chinai,[8] Michelle Emerson,[9] Kristin Beima-Sofie,[10] Emily Lawrence[1]

For numbered affiliations see end of article.

**Correspondence to**
Dr Bvudzai Priscilla Magadzire;
bvudzai.magadzire@villagereach.org

## ABSTRACT

**Objective** Immunisations are highly impactful, cost-effective public health interventions. However, substantial gaps in complete vaccination coverage persist. We aimed to describe caregivers' immunisation experiences and identify determinants of vaccine dropout.

**Design** We used a community-based participatory research approach employing Photovoice, SMS (short messaging service) exchanges and in-depth interviews. A team-based approach was used for thematic analysis. The Increasing Vaccination Model guided the analysis and identification of vaccination facilitators and barriers.

**Setting** This study was conducted in Zambézia province, Mozambique, in Namarroi and Gilé districts, where roughly 19% of children under 2 start but do not complete the recommended vaccination schedule.

**Participants** Participants were identified through health facility vaccination records and included caregivers of children aged 25–34 months who were fully vaccinated (n=10) and partially vaccinated (n=22). We also collected data from 12 health workers responsible for delivering immunisations at the selected health facilities.

**Results** Four main patterns of barriers leading to dropout emerged: (1) social norms and limited family support place the immunisation burden on mothers; (2) perceived poor quality of health services reduces caregivers' trust in vaccination services; (3) concern about side effects causes vaccine hesitancy; and (4) caregivers hesitate to seek and advocate for vaccination due to power imbalances with health workers. COVID-19 created additional barriers related to social distancing, mask requirements, supply chain challenges and disrupted outreach services. For most caregivers, dropout becomes increasingly likely with compounding barriers. Caregivers of fully-vaccinated children noted facilitators, including accompaniment to health facilities or assistance caring for other children, which enabled them to complete vaccination.

**Conclusions** Overcoming immunisation barriers requires strengthening health systems, including improving logistics to avert vaccine stockouts and building health worker capacity, including empathic communication with

## Strengths and limitations of this study

► Photovoice methods allowed participants to visually share their stories and perspectives, generating rich, detailed and nuanced information about the barriers that they experience in fully immunising their children.

► The community-based participatory research approach allowed community members to engage throughout the research stages, ensuring that the findings and resulting recommendations are community-centred.

► Due to the contextual nature of vaccination systems and vaccination barriers, the results may have limited generalisability beyond similar settings and populations.

► The COVID-19 pandemic prevented us from convening any focus groups and may have affected the quality of our participatory analysis activities, which had to be conducted virtually rather than in person.

caregivers. Consistent and reliable immunisation services could address access challenges and improve immunisation uptake, particularly in distant communities.

## INTRODUCTION

Childhood immunisations are highly impactful and cost-effective public health interventions, estimated to annually save 2–3 million lives worldwide.[1] In 1974, the World Health Organisation (WHO) established the Expanded Programme on Immunisation (EPI) to guide the development and implementation of vaccination programmes globally. Since then, coverage of the initial vaccines (protecting against diphtheria, tetanus, pertussis, measles, poliomyelitis and tuberculosis) has increased from fewer

than 5% of children in low-income and middle-income countries (LMICs) to more than 85%.[2 3] However, significant gaps in vaccination coverage remain. In 2019, an estimated 14 million children were unimmunised, and 5.7 million were under-immunised,[4] predominantly from lower socioeconomic and rural populations in LMICs.[5] The COVID-19 pandemic has reversed recent progress, with an estimated 23 million children not completing basic vaccinations in 2020.[6]

The Mozambique EPI, launched in 1979, provides immunisation services free of charge. Public sector health workers deliver EPI vaccines at fixed health posts, through routine mobile outreach activities and sometimes during campaigns for specific vaccines.[7] Since the early 2000's, Mozambique has implemented several strategies, including the WHO/United Nations Children's Fund (UNICEF) Reaching Every District/Reaching Every Community (RED/REC) strategy, to improve vaccination outreach in remote communities.[7] According to the Mozambique EPI manual, mobile vaccination brigades should provide services quarterly to communities residing further than 8 km from a health facility.[8] These outreach events are important for facilitating access to vaccines, as roughly 65% of the country's population lives in rural areas.[9] However, country-wide implementation of immunisation outreach strategies has been impeded by insufficient funding and limited material and human resources, resulting in coverage gaps.[7]

Routine vaccination coverage in Mozambique increased dramatically from 47% in 1997 to 63% in 2003, but progress has since slowed; according to the 2015 Survey of Indicators on Immunisation, Malaria and HIV/AIDS in Mozambique, only 66% of children were fully vaccinated,[10] and in 2019, UNICEF and WHO estimated the third dose of the diptheria-tetanus-pertussis vaccine (a common proxy for complete immunisation coverage) at 88%.[1] Within Mozambique, Zambézia province has the lowest vaccination coverage; in 2015, 50% of children under 2 had received the first doses of recommended immunisations, and 38% had started but dropped out from the vaccination schedule.[10] Low coverage of the first set of recommended vaccines indicates poor access to services or lack of vaccination acceptance, while high dropout rates suggest health system failures to successfully deliver repeated doses.[11] Low dropout rates are critical to preventing morbidity and mortality from vaccine-preventable diseases.[12] In the context of Zambézia province, this is particularly critical as there have recently been reported outbreaks of vaccine-preventable diseases such as cholera, polio and measles.[13] In Mozambique, there is a dearth of knowledge on drivers of routine immunisation dropouts specifically, as opposed to children who do not receive *any* of the routine immunisations. Determinants of immunisation dropout have been shown to be highly contextual, including combinations of individual, interpersonal and health systems factors.[14] We conducted interviews with caregivers of young children and health workers, using a community-based participatory research

(CBPR) approach, to identify key influences on under-2 immunisation dropout in two districts in Zambézia province, Mozambique.

## METHODS
### Study design
We conducted a qualitative study using CBPR and Photovoice methods. CBPR engages community representatives throughout the research process, reducing power imbalances between researchers and participants, creating an environment in which participants feel comfortable discussing sensitive topics and facilitating the co-creation of contextually-sensitive and community-centred knowledge.[15] Four Mozambican caregivers (referred to as Caregiver Researchers) were recruited from the local communities, were trained in research ethics and qualitative methods, led data collection and assisted with data analysis.

The Increasing Vaccination Model, developed by the WHO Behavioural and Social Drivers of Vaccination (BeSD) working group,[16] guided data collection and analysis. We adapted this model to incorporate elements of the UNICEF Caregiver Journey Model, dividing the 'practical issues' category into three subcategories to capture the vaccination timespan (pre-service, during-service and post-service) (online supplemental appendix A).

### Study setting, population and recruitment
The study was conducted between February 2020 and March 2021 in Gilé and Namarroi districts, Zambézia province. Zambézia, the second-most populated province in Mozambique, has many hard-to-reach communities with some of the lowest health outcomes; a child born in Zambézia is twice as likely to die before age five than a child born in the capital city of Maputo.[7 17] Of the province's 16 districts, Namarroi and Gilé have the highest under-immunisation rates, both at roughly 19%.[18] Gilé district has 11 fixed health facilities, serving a population of roughly 205 000.[18] Namarroi district has 10 fixed health facilities, serving a population of approximately 155 000.[18]

The study population included caregivers of fully-vaccinated or partially-vaccinated children who lived in catchment areas of the selected health facilities, and health workers who administer immunisations at those facilities. 'Caregivers', defined as parents or other guardians who take primary healthcare-seeking responsibility for the child, were eligible to participate if their child was 25–34 months old, an age by which the child should have received all routine childhood immunisations according to the Mozambique immunisation schedule (online supplemental appendix B). Caregivers were considered to be in the fully-vaccinated group (FV caregivers) if their child had received all 15 recommended immunisations, or the partially-vaccinated group (PV caregivers) if their child had received at least one but not all recommended immunisations. FV caregivers were included in the study to allow for comparison of experiences and barriers

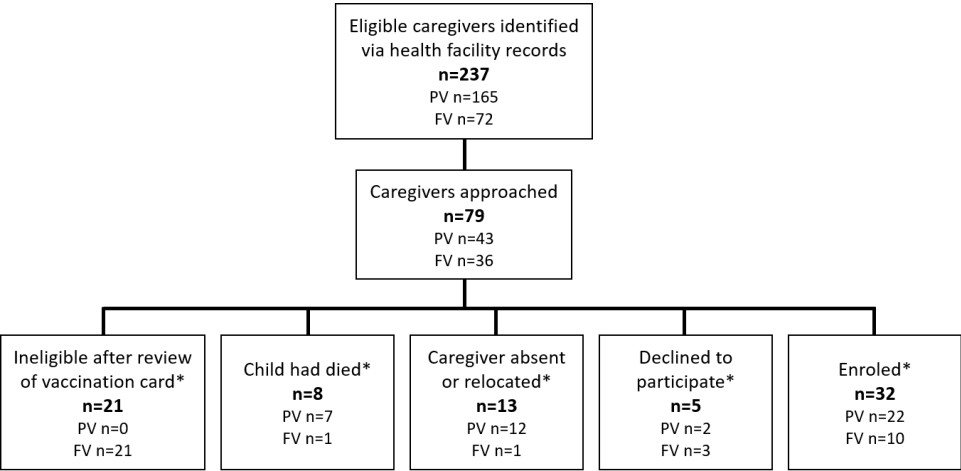

*Vaccination status reflects immunisation card, when available

**Figure 1** Recruitment and enrolment of caregiver participants. FV, fully vaccinated; PV, partially vaccinated.

between PV and FV caregivers. Health workers were eligible if they worked at the selected health facilities and if they were responsible for administering vaccinations to children under 2. Health workers were included in the study to provide additional perspective and nuance on the barriers that caregivers described.

We purposively selected eight health facilities (four per district) as recruitment sites to achieve a mixture of rural and peri-urban environments. Eligible caregivers were identified based on the facilities' immunisation records. Potential caregiver participants were excluded if they resided outside the health facility catchment area or their vaccination status did not meet definitions for partial or full vaccination. Caregiver Researchers then used a convenience sampling approach to visit caregivers' homes until they reached targeted enrolment numbers of approximately 20 PV caregivers and 10 FV caregivers. During recruitment, we verified vaccination status via home-based vaccination cards, which were considered more accurate when there were discrepancies with health facility records. Sixteen caregivers from each district were enrolled (figure 1), whereupon data saturation was reached. The study team purposively recruited and enrolled six health workers from each district, including at least one from each of the selected health facilities. No health workers declined to participate.

### Data collection
Caregiver Researchers were recruited via an open job posting advertised in Namarroi and Gilé districts. Applicants were required to have a 24–36 month-old child, and women were prioritised, as they were more representative of the study population. Three women from each district were selected based on these requirements and on their level of education, fluency in local languages and prior research experience. They participated in a 5-day in-person training on qualitative research methods, and contracts were extended to the two women from each district with the best performance during the training.

Guides for caregiver and healthcare worker semi-structured interviews were developed in English based on the Increasing Vaccination Model and translated into Portuguese (online supplemental appendices C and D). The guides were piloted with the Caregiver Researchers and one health worker, resulting in minor modifications to improve question clarity. Caregiver Researchers provided participants with study information. Basic demographic information was collected from all participants prior to each interview. Illiterate participants provided thumb-prints instead of signatures, otherwise written consent was obtained, including for use of photographs of them and their children.

Photovoice and semi-structured interviews were used to document caregivers' experiences with the under-2 immunisation process and to identify immunisation completion barriers and facilitators. Photovoice, a method through which people can share their stories using photographs that visually represent their experiences and perspectives, can empower participants to craft rich stories, generating data that might not be captured through standard interview techniques.[19 20] Caregiver Researchers provided cameras to caregiver participants, gave detailed instructions on their use (online supplemental appendix E), and asked caregivers to take photos related to their experiences immunising their child. Caregiver Researchers returned after 2–6 days and conducted audio recorded, in-depth interviews. Each caregiver selected the five photos that they felt represented the most meaningful depictions relating to their immunisation experience and began the interview by explaining the meaning and importance of each photo. Photo explanations were followed by a semi-structured set of questions for all caregivers to gather additional data. Initial interviews and transcripts were reviewed by EL and JP, who provided feedback and additional training to improve quality of subsequent interviews. Interviews were conducted in the local language of Elomué and lasted an

average of 38 min. Following each interview, Caregiver Researchers completed written interview debriefs, noting key insights. Audio recorded caregiver interviews were transcribed and translated from Elomué to Portuguese by Caregiver Researchers, audited for quality by study team members and then translated into English by an external translator and ME.

SMS (short messaging service) exchanges and semistructured interviews were used to document health workers' perceptions of caregiver immunisation experiences, their beliefs about causes of dropout and their experiences administering vaccines to children under 2. SMS exchanges took place during January and February 2021. Each health worker was provided a phone with the Telegram application set up for group chats with the Caregiver Researchers and study team members. Over a 3-week period, health workers were asked to send photos and messages whenever they had an experience or observation related to under-2 immunisation. Caregiver Researchers replied with follow-up probes to collect additional information. If more than 2 days elapsed without any messages exchanged, the Caregiver Researchers sent follow-up reminders. On average, each health worker sent 33 messages (range 11–51 messages). In February 2021, Caregiver Researchers conducted audio recorded interviews by phone with health workers in Portuguese, lasting an average of 70 min. Health worker interview audio recordings and SMS exchanges were transcribed and translated from Portuguese to English by an external translator.

## Data analysis

A team-based approach was used to perform a thematic analysis.[21] An initial codebook was developed deductively based on the Increasing Vaccination Model. Additional codes were added inductively following review of the first six transcripts from caregiver interviews, and the codebook was reviewed and updated by the Caregiver Researchers prior to coding. JP and EL individually coded the first three caregiver interview transcripts in ATLAS.ti V.9, and BPM reviewed the coded transcripts for intercoder agreement. JP and EL then divided and independently coded the remaining caregiver interview transcripts and summarised the main findings by vaccination status. On review of the first three transcripts from health worker SMS exchanges and interviews, additional codes were added to the codebook to capture the unique perspectives and experiences described by health workers. Health worker SMS exchanges and interview transcripts were coded by JP and EL and findings were summarised by JP, EL, BB, LB, AI, LCJ and ZM.

Caregiver Researchers participated in a virtual 2-week participatory analysis process facilitated by Portuguese-speaking study team members (AI and ME), in which they reviewed code summaries and representative quotations, identified themes and key barriers to vaccination and drew comparisons between PV and FV caregivers and between caregiver and health worker perspectives. An online data collaboration platform (Miro, 2021) was used to facilitate collaboration and establish consensus on themes.

The Photovoice photographs provided additional insight on what caregivers believed were the most significant aspects of their vaccination journeys. Each photo selected by the caregivers (including 110 photos from PV caregivers and 50 photos from FV caregivers) was reviewed and ccoded according to the subject of the photo, and the most common photo subjects of PV and FV caregivers' photos were compared. JP and EL reviewed the themes generated in the participatory analysis workshop and the photos to identify dropout patterns related to the Increasing Vaccination Model.

## Patient and public involvement

The Caregiver Researchers, as representatives of the study population, participated in data collection and analysis. Study results were disseminated to most participants during follow-up workshops in April and May 2021. Participants will be involved in dissemination of results back to their communities and to local health authorities.

## RESULTS

### Participants

Thirty-two caregivers participated in the study, and all were the mothers of the children. The majority lived in catchment areas of rural health facilities. On average, PV caregivers were older and less educated than FV caregivers (table 1). PV children were missing an average of 5.1 vaccines (range 1–14 missing vaccines), with the most commonly missed vaccines being Inactivated Polio and Measles-Rubella (tables 1 and 2).

Twelve health workers participated in the study (table 3). Most were men and had completed secondary education. Health workers had worked at the selected

| Table 1 Characteristics of caregiver participants and their children | | |
|---|---|---|
| | **PV Group (n=22) N (%) or median (IQR)** | **FV Group (n=10) N (%) or median (IQR)** |
| **District** | | |
| Namarroi | 11 (50) | 5 (50) |
| Gilé | 11 (50) | 5 (50) |
| **Health facility geography** | | |
| Rural | 16 (73) | 7 (70) |
| Peri-urban | 6 (27) | 3 (30) |
| Caregiver age (years) | 26 (22–30) | 21 (20–21.8) |
| **Caregiver education** | | |
| Some primary | 11 (50) | 3 (30) |
| Completed primary | 11 (50) | 7 (70) |
| Child age (months) | 30 (30–32) | 31.5 (29.3–32) |
| Child sex: female | 9 (41) | 6 (60) |
| Number of missing vaccines (of 15 possible) | 3.5 (2–8.8) | 0 |

FV, fully vaccinated; PV, partially vaccinated.

**Table 2** Vaccination completion of the partially-vaccinated children of caregiver participants

| Vaccine | Dose | Number (%) of partially-vaccinated children who received the dose (n=22)* |
|---|---|---|
| Tuberculosis | BCG | 22 (100) |
| Oral polio | OPV 0 | 15 (68.2) |
| | OPV 1 | 19 (86.36) |
| | OPV 2 | 15 (68.2) |
| | OPV 3 | 12 (54.6) |
| Pentavalent: diphtheria, pertussis, tetanus, hepatitis B, haemophilus influenzae type B | DPT-HepB-Hib 1 | 20 (90.1) |
| | DPT-HepB-Hib 2 | 15 (68.1) |
| | DPT-HepB-Hib 3 | 13 (59.1) |
| Pneumococcal conjugate | PCV 1 | 21 (95.45) |
| | PCV 2 | 17 (77.3) |
| | PCV 3 | 12 (54.6) |
| Rotavirus | RV 1 | 17 (77.3) |
| | RV 2 | 12 (54.6) |
| Inactivated polio | IPV | 7 (31.8) |
| Measles, rubella | MR 1 | 3 (13.6) |
| | MR 2† | 3 (13.6) |

*Note: Vaccination status is according to vaccination cards, or health facility records when no cards were available. Twins were counted as a single child.
†Note: The second dose of MR was added to the Mozambique expanded programme on immunisation schedule in November 2017. Due to the timing of the study, completion of MR 2 was not considered when determining vaccination status.
DPT, diptheria-tetanus-pertussis ; IPV, inactivated polio vaccine; MR, measles and rubella; OPV, oral polio vaccine; PCV, pneumococcal conjugate vaccine; RV, rotavirus.

**Table 3** Characteristics of health worker participants

| | N (%) or median (IQR) |
|---|---|
| **District** | |
| Namarroi | 6 (50) |
| Gilé | 6 (50) |
| **Education Level** | |
| Completed secondary | 11 (92) |
| Completed primary | 1 (8) |
| **Work experience** | |
| Total years of work experience | 7 (4.5–9.3) |
| Years at selected health facility | 4 (2–5) |
| **Job title** | |
| Preventive medical technician | 6 (50) |
| General nurse | 4 (33) |
| Maternal and child health nurse | 1 (8) |
| Nutrition technician | 1 (8) |
| **Immunisation-specific trainings** | |
| Attended training <5 years ago | 5 (42) |
| Attended training 5+ years ago | 2 (17) |
| Attended only informal vaccine discussions | 2 (17) |
| Never participated in trainings | 3 (25) |
| Sex: male | 9 (75) |

health facilities for an average of 4.7 years. All had children of their own, including five with children under 2.

### Photovoice findings

Photovoice outputs depicted the people, places and things that caregivers felt were most important in their vaccination journeys. There were several notable differences between the most common subjects of PV and FV caregivers' photos. Sixty-four per cent of PV caregivers, versus only 30% of FV caregivers, selected at least one photo depicting them caring for their child who was experiencing side effects after the vaccination, and many voiced concerns about these side effects. Conversely, 60% of FV caregivers selected at least one photo of their own child or other children in the community who were healthy and growing well due to vaccines, a subject that only 27% of PV caregivers photographed. Roughly half of caregivers in both groups took photos of family and friends, but they described those photos differently; FV caregivers were much more likely to describe active support (e.g., family accompanying them to the health facility), while PV caregivers typically described only passive encouragement from family or talked about times when family members did not provide help with the vaccination process. Roughly half of caregivers in both groups also took photos of bathing the child before vaccination and of walking to the health facility while carrying the child.

| Table 4 | Cross-domain influences on vaccine dropout | | |
|---|---|---|---|
| **Patterns of barriers and facilitators to vaccination completion** | **Increasing Vaccination Model categories** | **Specific barriers and facilitators** | **Illustrative quotation** |
| Social norms and limited family support place the burden of vaccination on mothers, compounding the challenges of accessing vaccination services. | ► Social processes<br>► Practical issues: Preparation | ► Long distance to health facility<br>► Physically challenging journey to vaccinate<br>► Caregiver illness or injury<br>► Family support<br>► Women's role in society | "I came back from the hospital and sat here on the mat… My leg was in a lot of pain again on my way to the hospital and I ended up not vaccinating [my child]. I'm not going there anymore. When I ask my husband to accompany our child to the hospital, he says that he is not able to take the child on his back." – PV Caregiver from Namarroi |
| Perceived poor quality of health services reduces caregivers' trust in the health system. | ► Motivation<br>► Practical issues: Preparation<br>► Practical issues: Point of care | ► Caregiver dissatisfaction with health services<br>► Vaccine stockouts<br>► Lack of healthcare workers | "Sometimes when we arrived at the hospital, they claimed that there were no vaccines, so the child was just weighed. When that happened, I'd regret the distance I had to travel. The following month I would not return with fear of spending money and not finding the vaccine." – PV Caregiver from Gilé |
| Concern about side effects, exacerbated by 'accumulation' of vaccines, leads to hesitancy. | ► What people think and feel<br>► Practical issues: Point of care<br>► Practical issues: After care | ► Fear of vaccine 'accumulation'<br>► Lack of information about vaccine side effects and schedule<br>► Perception that side effects are normal | "The child caught fevers [after vaccination]… Sometimes she had two or three days with fevers and swollen legs. She got more fevers in the months that [she] had multiple vaccines at the same time. As I was not informed about the side effects of getting different vaccines at the same time, the child would get home with many fevers and worry me." – PV Caregiver from Gilé |
| Power dynamics at the health facility make caregivers hesitant to seek or advocate for vaccination. | ► Social processes<br>► Motivation<br>► Practical issues: Preparation<br>► Practical issues: Point of care | ► Out-of-schedule vaccinations<br>► Hygiene<br>► Vaccination cards<br>► Vaccination session hours | "Whenever I take my child to the hospital to vaccinate, I first bathe [my child] to avoid rejection and other illnesses that can be caught due to poor hygiene. Since when you take the child without bathing, [the child] will not be vaccinated, it is refused. When taking the child without bathing, I am afraid that other mothers will laugh and consider me as a dirty person." – PV Caregiver from Gilé |

PV, partially vaccinated .

### Key influences on vaccine dropout

Vaccination barriers arose in all categories of the Increasing Vaccination Model. In general, PV caregivers described abandoning vaccination after encountering multiple cross-domain barriers, rather than dropout being caused by a single barrier.

> …my father's illness until death, in addition to my illness, money and not even someone to help me, these are the events that happened during the first 2 years of my son's life. – PV Caregiver from Gilé

Our analysis of caregiver experiences and photos, coupled with insight from health workers, revealed four cross-domain patterns of barriers and facilitators to vaccination completion. These patterns related to, (1) social norms and limited family support; (2) perceived poor quality of health services; (3) concern about side effects; and (4) power imbalances between caregivers and health workers (table 4). COVID-19 created additional barriers to vaccination completion. Unless otherwise indicated, the findings described below relate to both PV and FV caregivers.

### Social norms and limited family support place the burden of vaccination on mothers, compounding the challenges of accessing vaccination services

For many caregivers, travelling to a health facility required significant investments of time, effort and money. Some caregivers described selling crops or buying less food to save money for transport to the health facility. When they had no money for transportation, caregivers described having to walk for several hours while carrying their children, often through challenging and dangerous terrain and inclement weather, even spending the night because the trip was too long to complete in a single day.

> The hospital is very distant. On the days we have to go on foot, we come home with our feet swollen from so much walking. There are times when you cannot go home because it is late, and you are forced to sleep

there, even with nowhere to stay. – PV Caregiver from Gilé

PV caregivers noted that given the long distances required to travel to receive vaccinations, lack of support from other family members was a major barrier to completing the vaccination schedule. Limited family support was in part due to community beliefs that vaccination responsibility resides with mothers and that fathers should not be involved in the process. This belief resulted in some mothers abandoning the process when they were unable to travel to the facility due to illness or injury, and their husbands and other family members did not help bring the children to the health facility. Several other mothers reported dropping out due to lack of support with managing other conflicting obligations, such as caring for other children.

[My aunt] had a toothache and couldn't stay with [my other children]. When she told me this, I was left with no way out; I couldn't go to the hospital and leave my children alone. – PV Caregiver from Gilé

Social support from family and friends helped FV caregivers to overcome vaccination barriers. FV caregivers described receiving support in a variety of ways including reminders from family members, money for transport to the health facility, accompaniment to the facility by other caregivers or a family member, help with other household responsibilities and support in taking the child to the facility when the mother is sick.

Both caregivers and health workers noted that reliable, consistent mobile brigades would help to address the practical barriers associated with travelling to health facilities, especially for those mothers who lack support from family members. Several caregivers mentioned that mobile brigades had previously enabled them to complete vaccinations of older children, but that the brigades had become infrequent and unreliable. Health workers described barriers to running mobile brigades, including insufficient fuel and lack of staffing to manage both fixed posts and outreach services.

We need motorcycles and fuel so that we can run mobile brigades in the communities.… There are times when this doesn't happen 100% because the bikes are damaged, and we can't do maintenance.… When we do not have funds for fuel, sometimes the mobile brigades are paralyzed. – HW from Namarroi

### Perceived poor quality of health services reduces caregivers' trust in the health system

Many caregivers began the vaccination process highly motivated, and some described having positive experiences at health facilities. However, other caregivers were dissatisfied with their experiences at the service delivery point due to vaccine stockouts, long wait times and inconsistent health facility hours. PV caregivers also described being frustrated by interactions with disengaged health workers. After expending significant effort to reach the health facility, some caregivers arrived to find that health workers were late or were busy talking on their phones, causing them to lose faith in the health system and lose motivation to vaccinate.

I arrived at the hospital at a good time, before the activities started. I was waiting for the nurses until the time they arrived.… I was very indignant, since the nurses attended a person and interrupted to answer telephone calls that lasted many hours without consideration for mothers who live very far from the hospital. – PV Caregiver from Gilé

These dissatisfying experiences extended to general health services too; several PV caregivers discussed visits for treatment of malaria or other illnesses in which they received only partial doses of medications, leading them to also question the quality of vaccination services and abandon the vaccination process, rather than continue to invest time, effort and money.

Health workers described their own challenges in delivering vaccines and providing quality care to every child, including stockouts of vaccines and other supplies and high work burden due to insufficient human resources. Many health workers described vaccine stockouts lasting weeks to months with no guarantee of when doses would be available again. Several health workers raised concerns that aligned with those expressed by caregivers: that caregivers would not return after encountering stockouts due to distances and other difficulties associated with vaccine care seeking.

The whole province… has been out of stock of [the polio vaccine] for 2 months.… Considering that the vaccination program has a certain calendar, it is always embarrassing that this happens, because the children who should have had vaccinations in that period do not get them. – HW from Gilé

### Concern about side effects, exacerbated by 'accumulation' of vaccines, leads to hesitancy

Some caregivers understood that side effects such as swelling at the injection site or fever are common and should not be a reason to stop vaccination. However, many PV caregivers expressed concerns about these reactions, especially when they lasted for several days. This concern was also greater for PV caregivers who lived far away and felt that the long walk home aggravated the side effects. Some PV caregivers worried that 'simultaneous' or 'accumulated' vaccines—those given to catch up on missed doses—would exacerbate the fevers and swelling, causing some to drop out of the vaccination process.

…what made me abandon it were the side effects of the vaccine. I wanted to protect my son by abandoning the service and I didn't know I was harming the child… On the days that he received the vaccines simultaneously, he would have many fevers. This

 

happened when he spent many months without vaccinating. – PV Caregiver from Gilé

Health workers recognised these concerns. They felt that caregivers might not be aware that reactions are relatively common and are rarely harmful in the long-term, and they noted that better communication with caregivers is important for reducing dropouts caused by fear of side effects.

[The health worker] needs to sensitize the mother… why the child is getting that vaccine, and what will be the effect of that vaccine. For example, if the child has a fever after that vaccination, you tell the mother not to worry because it is normal… – HW from Namarroi

### Power dynamics at the health facility make caregivers hesitant to seek and advocate for vaccination services

Several caregivers described negative interactions with health workers. Caregivers feared that health workers would scold them and send them home if they came out of schedule after missed doses, did not bathe their child prior to the visit or forgot to bring their vaccination card.

If you have too many absences, when you arrive at the hospital you are insulted for not taking the child for the vaccination, even though you try to explain that it was not of your own will but because of the distance. – PV Caregiver from Gilé

There were strong social norms around bathing the child before vaccination to avoid being rejected or humiliated. PV caregivers' fear of humiliation was exacerbated by the public nature of the health facility, where immunisation activities and conversations take place in front of other caregivers due to lack of private rooms.

Caregivers frequently discussed the importance of vaccination cards. Both PV and FV caregivers felt that cards were important for tracking the vaccination schedule and for allowing health workers to easily identify their child's next vaccine. However, PV caregivers also noted how cards could negatively influence treatment and services, describing fears around being verbally abused and rejected by health workers over lost or damaged cards. Caregiver mothers who had given birth outside of a facility saw the cards as prohibitive of receiving vaccination services, fearing that if they requested a vaccination card, health workers would admonish them for the non-institutional delivery.

[My child] didn't get all the vaccinations because she didn't have a card. When it got torn, I arrived at the hospital to ask for a new card. I was told to get the previous one and I was left with no options. – PV Caregiver from Namarroi

PV caregivers noted that power dynamics sometimes also resulted in missed vaccination opportunities. A few PV mothers described instances when health workers sent them home without delivering any vaccinations. Even when PV caregivers believed that their children needed to be vaccinated, they were hesitant to approach the health workers to request the vaccine, fearing that they would be yelled at.

Many times when I went to the hospital, they just sent my daughters to weigh and said they had already completed the vaccinations and didn't even check the children's cards. But as a mother, I knew that the last vaccine was missing and I couldn't question it for fear of being humiliated. – PV Caregiver from Gilé

From the health worker perspective, however, some of these missed opportunities were due to challenges that they faced regarding vaccine delivery and cold chain policies. Some health workers said they do not vaccinate children over the age of 2, and one described policies that limit them to opening vaccine freezers just once at the start of the vaccination session and once at the end. If caregivers arrived after the vaccines were put away, the health workers could not retrieve the vaccines. Additional missed opportunities to vaccinate occurred at two health facilities; at one, there was no maternal and child health nurse to manage vaccination of newborns on weekends, and at the other, the maternal and child health nurses had been trained in vaccination but were not administering them to newborns.

All deliveries that take place on the weekends must return on Monday to vaccinate. Sometimes there are missed opportunities because mothers live far away…. [Maternal and child health] nurses have training in vaccination, and it is possible to manage the vaccination of newborns—just apply the BCG vaccine and then dismiss the mothers—but this is not happening. – HW from Gilé

The data suggest a breakdown in communication between health workers and caregivers; health workers do not always communicate their logistical and resource constraints, and caregivers are afraid to ask about why their children are not receiving vaccines.

For a few months that I went to the hospital they just weighed the children and returned the card to me without saying anything, but I was afraid to ask. I was silent without knowing if it was a lack of vaccines at the hospital or bad luck for me. – PV Caregiver from Gilé

### Impacts of COVID-19 on vaccination experiences

Though some caregivers and health workers felt that COVID-19 had not impacted the vaccination process, others described changes to health services and other barriers to vaccination resulting from the pandemic. In some cases, caregivers were hesitant to go to the health facility due to fear of contracting COVID-19 or general discomfort around the new health facility policies. Mask requirements were another reported deterrent, as

caregivers who did not have their own masks knew that they would not be served.

> In the hospital, without the use of masks, people are not attended, as well as on public transport. Our habits of shaking hands and staying close to our friends is also forbidden. These are things that make us distressed and worried without knowing how long it will take before it passes. – PV Caregiver from Gilé

At the health facility level, health workers described shortages of general supplies, such as gloves, and increased vaccine stockouts.

> At times, we lacked some vaccines because…the borders closed…due to COVID-19.… For example, we are currently without BCG and polio vaccines.… There is difficulty in [transporting the vaccines to] the province. – HW from Namarroi

Health workers also mentioned challenges maintaining social distancing because the waiting rooms are too small to accommodate all the caregivers, forcing caregivers and their children to wait outside even in hot sun or rain.

Both caregivers and health workers noted that mobile brigades have been especially infrequent during the COVID-19 pandemic. Health workers described difficulties in organising mobile brigades due to social distancing requirements as well as lack of fuel and required supplies. Caregivers commented that they were unaware of brigade schedules due to closures of schools and churches, which previously were their main sources of information about brigades. When mobile brigades did happen, health workers reported that far fewer caregivers than usual brought their children due to fears of congregating in groups.

> Now we go to the community, and we find there are not enough children due to the pandemic, and messages are arriving in the communities saying that we cannot concentrate…50 people together.… If before COVID-19, I could reach 200 children in immunization, today when I go, I can only reach 30 to 50 children. – HW from Namarroi

## DISCUSSION

Our study highlighted the immense effort that caregivers put into vaccinating their children and the numerous and difficult barriers they face. Primary barriers identified in our study included: practical issues related to health facility access, unreliable and poorly perceived service quality, negative interactions with health workers, caregiver misunderstandings about vaccine side effects and unsupportive household dynamics. Our findings are consistent with many of the dropout determinants that have previously been identified in systematic reviews that included LMIC settings, including: access, health worker availability, missed opportunities, service reliability, family and gender dynamics, childcare challenges

for siblings, lack of motivation, fear of side effects, mistrust of the health system and health staff attitudes and behaviour.[14 22–25] Other studies in sub-Saharan Africa have identified specific vaccination barriers very similar to ours, including poor interactions with health workers, such as being verbally abused if the child did not look presentable, as well as incurring expenses to reach a facility only to find vaccine stockouts.[23] Our findings also align with results from cross-sectional studies in Mozambique, which found that dropout was related to lack of access to vaccination facilities, concern about receiving multiple vaccines at once, household decision-making processes and children born at home or outside of Mozambique.[23 26 27] Additional barriers that have been previously identified in Mozambique, but which did not arise explicitly in this study, were maternal education and poverty, though we did see that illiteracy and cost were barriers. This study also did not find evidence of vaccination completion relating to birth order or number, determinants that have been identified in other African countries.[28]

Our study generated several new findings related to caregivers' interactions with health workers and their perceptions of service quality, caregivers' hesitancy related to vaccine 'accumulation', the importance of family support and the shortcomings of the existing outreach services. Fear of humiliation or reprimand contributed to caregivers' perceptions of poor service and discouraged them from continuing with vaccination once their child had fallen off schedule. The perceived poor quality of *general* health services influenced some caregivers' decisions to stop seeking vaccination services. Communication breakdowns may have made caregivers unaware of health workers' constraints, exacerbating their perceptions of poor service quality. Many caregivers in our study were more fearful of side effects when they believed that their child was receiving too many vaccines simultaneously and especially in the context of catch-up administration. Our findings showed that family support of mothers is a strong enabler to overcoming many vaccination barriers, but the dominant social norms exempt husbands and other family members from assisting with the vaccination process. Finally, the data showed that currently, outreach services are not meeting the needs of caregivers in distant communities; mobile vaccination brigades have been infrequent, especially due to COVID-19, and caregivers do not always receive advance notice of them. Overall, vaccine dropout was related to compounding barriers that eventually tip the scales in favour of abandoning the vaccination cycle. This study has generated a more comprehensive understanding of how those barriers interact together.

### Implications on policy and practice
Health systems strengthening is receiving increasing attention globally.[29] Our findings support the need for this approach to increase vaccination completion, including improved supply chains to ensure adequate

stocks of vaccines and other supplies, sufficient health workforce and health facilities that allow for privacy and confidentiality. Capacity building of health workers, including training, retraining and supportive supervision is also needed.[30] These capacity building approaches should not only focus on technical content, but should support health workers to engage in respectful, empathic and patient-centred communication.[31] Specifically, better messaging is needed on vaccine availability, scheduling and hours of immunisation sessions, common reactions especially for off-schedule vaccines and policies around issuing new vaccination cards. Involvement of community leaders, religious leaders and community health workers in those communication strategies may improve vaccination uptake.[32] Ultimately, however, enhanced relationships with patients and the community also require a sufficient and motivated health workforce.[31]

Our findings suggest that reinvestment in community-centred vaccine outreach services could improve completion rates. Regularly scheduled outreach events have the potential to greatly facilitate access to immunisation services in distant communities, especially when communities are involved in planning them.[33] The WHO and UNICEF have endorsed the RED/REC strategy since 2002,[33 34] but our findings suggest that there has been inadequate resource allocation to implement RED/REC in the study communities.

To successfully implement these recommendations, there is a need for further research on the best channels of communication between health facilities and the communities they serve, as well as on how best to design and implement empathy-building trainings for health workers. Additionally, there is a need to better understand how to engage husbands and other family members in the vaccination process to reduce the burden on mothers.

Unique strengths of this study include the involvement of Caregiver Researchers, who drew from their own experiences and familiarity with the community and research topic during data collection and analysis, and the use of multiple data collection methods that generated rich visual, written and oral data. In addition, centring of the study around the Increasing Vaccination Model ensured that we holistically considered the different categories of vaccination barriers and facilitators.

Our study has several limitations. The specific barriers we identified may not be generalisable to other contexts because influences on vaccine dropout are highly contextual, and determinants relate to social norms, health service policies and geography. However, the complex nature of compounding barriers may translate to other settings. An additional limitation is that this study was conducted during the COVID-19 pandemic, when many health services were disrupted. While this provided a unique opportunity to examine barriers caused by this crisis, this also may have biased our findings about dropout during non-pandemic times and may have affected the quality of the participatory analysis process, which was conducted virtually. Finally, during recruitment, 13 caregivers were not enrolled due to absence or seasonal relocation to their farms. These caregivers were disproportionately caregivers of PV children, and failure to include them may have influenced the results. Despite these limitations, this study demonstrates that immunisation dropout is complex, with caregivers facing different combinations of barriers and facilitators over the course of their child's immunisation journey. The findings highlight the need for improved quality and reliability of vaccination services and for community-centred solutions that remove practical barriers and empower caregivers to complete the vaccination process.

**Author affiliations**
[1]Research, Evidence & Learning, VillageReach, Seattle, WA, USA
[2]Health Systems, VillageReach, Cape Town, South Africa
[3]Mozambique, VillageReach, Quelimane, Mozambique
[4]Department of Pediatrics, University of Washington, Seattle, Washington, USA
[5]Mozambique, VillageReach, Maputo, Mozambique
[6]Independent, Namarroi, Mozambique
[7]Independent, Gilé, Mozambique
[8]Zambézia Provincial Directorate of Health, Quelimane, Mozambique
[9]Health Systems, VillageReach, Seattle, Washington, USA
[10]Department of Global Health, University of Washington School of Public Health, Seattle, Washington, USA

**Contributors** JP, BPM, AD, BC, JEC and EL contributed to study design and logistics. AD, AI, BBLB, LCJ and ZM contributed to data collection. JP, BPM, DD, AI, BBLB, LCJ, ZM, ME, KB-S and EL contributed to data analysis. JP took the lead in writing the manuscript, with consultation from DD, KB-S and EL. All authors provided critical feedback and helped shape the final manuscript. EL is the guarantor of the manuscript.

**Funding** This work was supported by Wellcome Trust grant number 219237/Z/19/Z. The funder did not have a role in the study design, analysis, data interpretation or manuscript development.

**Competing interests** None declared.

**Patient consent for publication** Not applicable.

**Ethics approval** This study involves human participants and was approved by The Mozambique National Bioethics Committee for Health approved the study procedures (Ref: 259/CNBS/20) and the University of Washington Institutional Review Board exempted the study from review (STUDY00011999). Participants gave informed consent to participate in the study before taking part.

**Provenance and peer review** Not commissioned; externally peer reviewed.

**Data availability statement** No data are available. No additional data are available.

**ORCID iDs**
Jocelyn Powelson http://orcid.org/0000-0002-9935-9122
Bvudzai Priscilla Magazire http://orcid.org/0000-0002-4164-6233

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
