## [Reviewer comments · BMJ Open]

ARTICLE DETAILS

TITLE (PROVISIONAL)	Determinants of immunization dropout among children under the age of two in Zambézia Province, Mozambique: A community-based participatory research study using Photovoice
AUTHORS	Powelson, Jocelyn; Magadzire, Bvudzai; Draiva, Abel; Denno, Donna; Ibraimo, Abdul; Benate, Bonifácia; Jahar, Lídia; Marrune, Zélia; Chilundo, Baltazar; Chinai, Jalilo; Emerson, Michelle; Beima-Sofie, Kristin; Lawrence, Emily

VERSION 1 – REVIEW

REVIEWER	Jackson, Cath University of York, York Trials Unit
REVIEW RETURNED	26-Nov-2021

GENERAL COMMENTS	I found this to be a well written, interesting paper presenting a carefully designed and executed study. The findings have immediate relevance for addressing caregivers' multifaced barriers to vaccinating their children. I particularly liked the use of caregiver researchers, photos and SMS data and the use of the BeSD model as the theoretical underpinning of the study. My comments relate mainly to providing more detail in places to the methods you used and, more significantly to the presentation of your findings. I hope they are helpful. INTRODUCTION • Line 35 – do you have a more recent reference than 2015? POPULATION • It would be useful to add a sentence to say why you included FV parents. I assume it is to identify drivers and also to provide a comparison (you can then be sure which barriers are specific to PV and which exist for FV but hey overcome them)• Also why health workers? what does their perspective add?• How did you go from 237 to 79 caregivers? what was your process of selection – random? purposive (what criteria?) DATA COLLECTION • Did you make any changes to the interview guides after piloting? DATA ANALYSIS • How did you analyse the health worker interview and SMS data? Did you use the same code book and analyse with the caregiver data? or have a different codebook, do a separate analysis and then triangulate? RESULTS
---

	 • I find Table 4 to be somewhat unusual. It is a table of quotes which seems odd when you are putting quotes in the text as well. My suggestion would be to change Table 4 into a diagram presenting the 4 cross-domain influences and their sub-topics e.g. distance, illness and injury. Even better would be to somehow merge this information and structure into Appendix A. • You then need check that your text on the 4 headline influences covers all the sub-topics that were in the Table (I haven't checked but expect it already does) and decide which quotes from the table/text you want to keep – I doubt there is space to keep them all. • For me, this is an important change to make. • Can you add a sentence at the start of the findings to say that unless indicated, the findings for the caregivers relate to both PV and FV. At the moment, in places I am unsure when you say “many caregivers” “some caregivers” whether you mean both groups? DISCUSSION  • Line 33 – I think it is misunderstanding about vaccine side effects rather than a lack of awareness? • References 4 and 5 are too old, there is a lot more recent literature for you to consider. Take a look at the work by Kaufman et al 2021 https://gh.bmj.com/content/6/9/e006860 This is the most recent review there is and also the reference list may identify some reviews of studies in Africa? • Can you say a bit more about how your findings compare to the African literature? • Tell us about the strengths of your study – there are so many – using use of caregiver researchers (refer to the literature on “insider researchers”, using the photos and SMS data, the BeSD model ensures you think holistically about barriers/drivers and help develop evidence and theory-informed policy and interventions.
--	--

REVIEWER	Widayanti, Anna Wahyuni Gadjah Mada University, Pharmaceutics
REVIEW RETURNED	28-Nov-2021

GENERAL COMMENTS	Comments: The topic of the study is of interest, and it's also interesting to see that the authors also applied a photovoice methods. The article is also well written. I have few comments that may improve the quality of the article. Introduction: The authors mentioned: “Within Mozambique, Zambézia Province has the lowest vaccination coverage; in 2015, 50% of children under 2 had received the first doses of recommended immunizations, and 38% had started but dropped out from the vaccination schedule” also in the methods section: “Of the province’s 16 districts, Namarroi and Gilé have the highest under-immunization rates, both at roughly 19%.” From that, we can see that the coverage of first dose of vaccination or vaccination coverage was also very low. Could you explain why your research focused only on the factors that may influencing the dropout rate of vaccination? Methods: Table 1 is a bit confusing, as the authors also mentioned the partially vaccinated children who received the dose with n=22. When first read, I assume that the number of children population who partially vaccinated was only 22. It turns out that 22 was the number of participants involved in the PV groups. I suggest that the authors
--

	move the percentage to the results part. Methods and results: The authors used Photovoice to document caregivers' experiences with the under-2 immunization process and to identify immunization completion barriers and facilitators. However, I do not see the data from the photovoice methods in the results parts. How many photos collected from the respondents and what are the photos about? How the researchers analyze the photos to generate themes from the photos? The authors need to provide the results from photovoice methods in the results part. Is this method effective enough to get a more comprehensive stories from the participants?
--	---

VERSION 1 – AUTHOR RESPONSE

Reviewer 1: Dr. Cath Jackson, University of York

Comments: I found this to be a well written, interesting paper presenting a carefully designed and executed study. The findings have immediate relevance for addressing caregivers' multifaced barriers to vaccinating their children. I particularly liked the use of caregiver researchers, photos and SMS data and the use of the BeSD model as the theoretical underpinning of the study.

My comments relate mainly to providing more detail in places to the methods you used and, more significantly to the presentation of your findings. I hope they are helpful.

Specific Comments:

Introduction:

1) Line 35 – do you have a more recent reference than 2015?

We have added a 2019 estimate of DTP3 coverage from WHO/UNICEF. However, we have also retained the original coverage numbers from 2015, as that is the most recent high-quality data available. More recent data from Mozambique include many vaccination coverage rates that exceed 100% due to inaccuracies in the denominator resulting from poor population estimates. We are less concerned about drawing from the more recent data (2019) on under-immunization rates for Namarroi and Gilé districts in the methods section, as those values are calculated with a more accurate denominator. The text (clean copy - page 4, line 32; marked copy – page 24, line 32) now reads: "Routine vaccination coverage in Mozambique increased dramatically from 47% in 1997 to 63% in 2003, but progress has since slowed; according to the 2015 Survey of Indicators on Immunization, Malaria, and HIV/AIDS in Mozambique, only 66% of children were fully vaccinated,[10] and in 2019, UNICEF and WHO estimated the third dose of the Diphtheria-Tetanus-Pertussis (DPT 3) vaccine (a common proxy for complete immunization coverage) at 88%.[1]"

Population:

2) It would be useful to add a sentence to say why you included FV parents. I assume it is to identify drivers and also to provide a comparison (you can then be sure which barriers are specific to PV and which exist for FV but hey overcome them)

Thank you for the suggestion. We have added a sentence to explain that the FV caregivers were included to provide a point of comparison. The added sentence (clean copy - page 5, line 40; marked copy – page 25, line 42) reads:

"FV caregivers were included in the study to allow for comparison of experiences and barriers between PV and FV caregivers."

3) Also why health workers? what does their perspective add?

We added a sentence to explain that health workers were included in the study to provide additional perspective and nuance to the barriers that caregivers identify, including why caregivers might experience dissatisfying care or have negative interactions with health workers. The added sentence (clean copy - page 5, line 43; marked copy – page 25, line 45) reads:

“Health workers were included in the study to provide additional perspective and nuance on the barriers that caregivers described.”

4) How did you go from 237 to 79 caregivers? what was your process of selection – random? purposive (what criteria?)

We have edited to clarify that the initial list of 237 caregivers was restricted based on residing within the health facility catchment area and vaccination status, and that a convenience sampling approach was then used to arrive at the final sample, as caregiver researchers visited eligible caregivers until enrollment goals were met. The text (clean copy - page 5, line 48; marked copy – page 25, line 51) now reads:

“Potential caregiver participants were excluded if they resided outside the health facility catchment area or their vaccination status did not meet definitions for partial or full vaccination. Caregiver Researchers then used a convenience sampling approach to visit caregivers’ homes until they reached targeted enrollment numbers of approximately 20 PV caregivers and 10 FV caregivers.”

Data Collection

5) Did you make any changes to the interview guides after piloting?

We have added to the sentence to clarify that minor modifications were made to improve the clarity of the interview questions. The text (clean copy - page 6, line 19; marked copy – page 26, line 24) now reads:

“The guides were piloted with the Caregiver Researchers and one health worker, resulting in minor modifications to improve question clarity.”

Data Analysis

6) How did you analyse the health worker interview and SMS data? Did you use the same code book and analyse with the caregiver data? or have a different codebook, do a separate analysis and then triangulate?

We have added text to explain the analysis of health worker data. Health worker SMS and interview data was coded using the same codebook, with additional codes added that related specifically to health worker perspectives. Coded health worker data was evaluated alongside caregiver data, which fed into the participatory analysis workshop. The added text (clean copy - page 7, line 18; marked copy – page 27, line 24) reads:

“Upon review of the first three transcripts from health worker SMS exchanges and interviews, additional codes were added to the codebook to capture the unique perspectives and experiences described by health workers. Health worker SMS exchanges and interview transcripts were coded by JP and EL and findings were summarized by JP, EL, BB, AI, LJ, and ZM.”

Results

7) I find Table 4 to be somewhat unusual. It is a table of quotes which seems odd when you are putting quotes in the text as well. My suggestion would be to change Table 4 into a diagram presenting the 4 cross-domain influences and their sub-topics e.g. distance, illness and injury. Even better would be to somehow merge this information and structure into Appendix A.

You then need check that your text on the 4 headline influences covers all the sub-topics that were in the Table (I haven’t checked but expect it already does) and decide which quotes from the table/text you want to keep – I doubt there is space to keep them all.

For me, this is an important change to make.

We have revised Table 4 (clean copy - page 10; marked copy – page 30) so that instead of containing only representative quotations, it now includes the key barriers and facilitators, relevant domains of

the Increasing Vaccination Model, and a single representative quotation for each of the four dropout patterns. We have confirmed that the text in the results section covers all of the information that is presented in the table.

8) Can you add a sentence at the start of the findings to say that unless indicated, the findings for the caregivers relate to both PV and FV. At the moment, in places I am unsure when you say “many caregivers” “some caregivers” whether you mean both groups?

We have added the recommended sentence (clean copy - page 10, line 23; marked copy – page 30, line 31).

“Unless otherwise indicated, the findings described below relate to both PV and FV caregivers.”

Within the results section, we have also clarified throughout which caregivers are being referenced, and added additional text to better explain differences in findings on vaccination cards between PV and FV caregivers. The text relating to vaccination cards (clean copy - page 13, line 50; marked copy – page 34, line 3) now reads:

“Caregivers frequently discussed the importance of vaccination cards. Both PV and FV caregivers felt that cards were important for tracking the vaccination schedule and for allowing health workers to easily identify their child’s next vaccine. However, PV caregivers also noted how cards could negatively influence treatment and services, describing fears around being verbally abused and rejected by health workers over lost or damaged cards. Caregiver mothers who had given birth outside of a facility saw the cards as prohibitive of receiving vaccination services, fearing that if they requested a vaccination card, health workers would admonish them for the non-institutional delivery.”

Discussion

9) Line 33 – I think it is misunderstanding about vaccine side effects rather than a lack of awareness?

We agree and have changed the language accordingly. The text (clean copy - page 15, line 36; marked copy – page 35, line 45) now reads:

“Primary barriers identified in our study included: practical issues related to health facility access, unreliable and poorly perceived service quality, negative interactions with health workers, caregiver misunderstandings about vaccine side effects, and unsupportive household dynamics.”

10) References 4 and 5 are too old, there is a lot more recent literature for you to consider. Take a look at the work by Kaufman et al 2021 <https://gh.bmj.com/content/6/9/e006860> This is the most recent review there is and also the reference list may identify some reviews of studies in Africa?

Thank you for the suggested references. We have updated the literature review in this section and added additional relevant content from new sources.

We have added the following citations:

• Cobos Muñoz D, Monzón Llamas L, Bosch-Capblanch X. Exposing concerns about vaccination in low- and middle-income countries: a systematic review. *Int J Public Health* 2015;60:767–80.

doi:10.1007/s00038-015-0715-6

• Kaufman J, Tuckerman J, Bonner C, et al. Parent-level barriers to uptake of childhood vaccination: a global overview of systematic reviews. *BMJ Glob Health* 2021;6:e006860. doi:10.1136/bmjgh-2021-006860

• Merten S, Hilber AM, Biaggi C, et al. Gender Determinants of Vaccination Status in Children: Evidence from a Meta-Ethnographic Systematic Review. *PLOS ONE* 2015;10:e0135222.

doi:10.1371/journal.pone.0135222

The text (clean copy - page 15, line 40; marked copy – page 35, line 49) now reads:

“Our findings are consistent with many of the dropout determinants that have previously been identified in systematic reviews that included LMIC settings, including: access, health worker availability, missed opportunities, service reliability, family and gender dynamics, childcare challenges for siblings, lack of motivation, fear of side-effects, mistrust of the health system, and health staff attitudes and behavior.[14,22–25]”

11) Can you say a bit more about how your findings compare to the African literature?

We have added additional content and citations that relate specifically to vaccination barriers in African countries and have compared our findings with this literature.

The new citations include:

- Cobos Muñoz D, Monzón Llamas L, Bosch-Capblanch X. Exposing concerns about vaccination in low- and middle-income countries: a systematic review. *Int J Public Health* 2015;60:767–80. doi:10.1007/s00038-015-0715-6
- Tauli M de C, Sato APS, Waldman EA. Factors associated with incomplete or delayed vaccination across countries: A systematic review. *Vaccine* 2016;34:2635–43. doi:10.1016/j.vaccine.2016.04.016

The text (clean copy - page 15, line 45; marked copy – page 35, line 54) now reads:

“Other studies in sub-Saharan Africa have identified specific vaccination barriers very similar to ours, including poor interactions with health workers, such as being verbally abused if the child did not look presentable, as well as incurring expenses to reach a facility only to find vaccine stockouts.[23] Our findings also align with results from cross-sectional studies in Mozambique, which found that dropout was related to lack of access to vaccination facilities, concern about receiving multiple vaccines at once, household decision-making processes, and children born at home or outside of Mozambique.[23,26,27] Additional barriers that have been previously identified in Mozambique, but which did not arise explicitly in this study, were maternal education and poverty, though we did see that illiteracy and cost were barriers. This study also did not find evidence of vaccination completion relating birth order or number, determinants that have been identified in other African countries.[28]”

12) Tell us about the strengths of your study – there are so many – using use of caregiver researchers (refer to the literature on “insider researchers”, using the photos and SMS data, the BeSD model ensures you think holistically about barriers/drivers and help develop evidence and theory-informed policy and interventions.

We have added a paragraph about the strengths of the study (including the use of Caregiver Researchers, the multiple methods of data collection, and the BeSD model) and about how our methods and approach contributed to the findings. The new text (clean copy - page 17, line 6; marked copy – page 37, line 14) reads:

“Unique strengths of this study include the use of Caregiver Researchers, who drew from their own experiences and familiarity with the community and research topic during data collection and analysis, and the use of multiple data collection methods that generated rich visual, written, and oral data. In addition, centering of the study around the BeSD model ensured that we holistically considered the different categories of vaccination barriers and facilitators.”

Reviewer 2: Dr. Anna Wahyuni Widayanti, Gadjah Mada University

Comments: The topic of the study is of interest, and it's also interesting to see that the authors also applied a photovoice methods. The article is also well written.

I have few comments that may improve the quality of the article:

Introduction

1) The authors mentioned: “Within Mozambique, Zambézia Province has the lowest vaccination coverage; in 2015, 50% of children under 2 had received the first doses of recommended immunizations, and 38% had started but dropped out from the vaccination schedule” also in the methods section: “Of the province’s 16 districts, Namarroi and Gilé have the highest under-immunization rates, both at roughly 19%.”

From that, we can see that the coverage of first dose of vaccination or vaccination coverage was also very low. Could you explain why your research focused only on the factors that may influencing the dropout rate of vaccination?

We focused on the factors that may be influencing dropout because there is less existing literature that specifically focuses on dropout determinants as opposed to factors leading to non-immunization

or to both non-immunization and under-immunization more generally. As noted in the manuscript, the literature suggests that causes of non-immunization are different from causes of under-immunization, and we felt that there was a greater knowledge gap in the latter area. Additionally, in Zambézia Province, where this study took place, there have been recent outbreaks of vaccine-preventable diseases, which are likely a result of low coverage and dropout rates. The text (clean copy - page 4, line 43; marked copy – page 24, line 43) now reads:

“Low dropout rates are critical to preventing morbidity and mortality from vaccine-preventable diseases.[12] In the context of Zambézia Province, this is particularly critical as there have recently been reported outbreaks of vaccine-preventable diseases such as cholera, polio, and measles.[13] In Mozambique, there is a dearth of knowledge on drivers of routine immunization dropouts specifically, as opposed to children who do not receive any of the routine immunizations.”

Methods

2) Table 1 is a bit confusing, as the authors also mentioned the partially vaccinated children who received the dose with n=22. When first read, I assume that the number of children population who partially vaccinated was only 22. It turns out that 22 was the number of participants involved in the PV groups. I suggest that the authors move the percentage to the results part.

We understand that the table could be misleading and have edited it to be clearer. The vaccination schedule, referenced in the methods section, has been removed from the table and is now in Appendix B. The vaccination rates of the children of PV caregiver participants are now presented in the results section in Table 2 (clean copy – page 8; marked copy - page 28).

Methods and results

3) The authors used Photovoice to document caregivers’ experiences with the under-2 immunization process and to identify immunization completion barriers and facilitators. However, I do not see the data from the photovoice methods in the results parts. How many photos collected from the respondents and what are the photos about? How the researchers analyze the photos to generate themes from the photos? The authors need to provide the results from photovoice methods in the results part. Is this method effective enough to get a more comprehensive stories from the participants?

We have added 1-2 sentences each in the data collection and data analysis sections to better explain the Photovoice approach and how the photos were used and a new section to the results, where we present the Photovoice findings. New text in the methods section now reads:

Data Collection (clean copy - page 6, line 35; marked copy – page 26, line 41):

“Each caregiver selected the five photos that they felt were the most meaningful depictions relating to their immunization experience and began the interview by explaining the meaning and importance of each photo. Photo explanations were followed by a semi-structured set of questions for all caregivers to gather additional data.”

Data Analysis (clean copy - page 7, line 33; marked copy – page 27, line 40):

“Each photo selected by the caregivers (including 110 photos from PV caregivers and 50 photos from FV caregivers) was reviewed and classified according to the subject of the photo, and the most common photo subjects of PV and FV caregivers’ photos were compared.”

New text in the results section (clean copy - page 9, line 44; marked copy – page 29, line 52) reads:

“Photovoice outputs depicted the people, places, and things that caregivers felt were most important in their vaccination journeys. There were several notable differences between the most common subjects of PV and FV caregivers’ photos. Sixty-four percent of PV caregivers, versus only 30% of FV caregivers, selected at least one photo depicting them caring for their child who was experiencing side-effects after the vaccination, and many voiced concerns about these side-effects. Conversely, 60% of FV caregivers selected at least one photo of their own child or other children in the community who were healthy and growing well due to vaccines, a subject that only 27% of PV caregivers photographed. Roughly half of caregivers in both groups took photos of family and friends, but they described those photos differently; FV caregivers were much more likely to describe active support

(e.g. family accompanying them to the health facility), while PV caregivers typically described only passive encouragement from family or talked about times when family members did not provide help with the vaccination process. Roughly half of caregivers in both groups also took photos of bathing the child before vaccination and of walking to the health facility while carrying the child.”

VERSION 2 – REVIEW

REVIEWER	Jackson, Cath University of York, York Trials Unit
REVIEW RETURNED	14-Jan-2022
GENERAL COMMENTS	Thanks for making the revisions so carefully, I think the paper looks great now!